# Grid-Characteristic Method on Overlapping Curvilinear Meshes for Modeling Elastic Waves Scattering on Geological Fractures

**Nikolay I. Khokhlov** [1,2,*] , **Alena Favorskaya** [1,2] **and Vladislav Furgailo** [1]

1   Moscow Institute of Physics and Technology, 9 Institutskiy Lane, Dolgoprudny 141701, Russia
2   Scientific Research Institute for System Analysis of the Russian Academy of Sciences, Moscow 117218, Russia
*   Correspondence: khokhlov.ni@mipt.ru

**Abstract:** Interest in computational methods for calculating wave scattering from fractured geological clusters is due to their application in processing and interpreting the data obtained during seismic prospecting of hydrocarbon and other mineral deposits. In real calculations, numerical methods on structured, regular (Cartesian) computational grids are used to conserve computational resources though these methods do not correctly model the scattering of elastic waves from fractures that are not co-directed to the coordinate axes. The use of computational methods on other types of grids requires an increase in computational resources, which is unacceptable for the subsequent solution of inverse problems. This article is devoted to a possible solution to this problem. We suggest a novel modification of a computational grid-characteristic method on overlapping curvilinear grids. In the proposed approach, a small overlapping curvilinear grid is placed around a fracture that smoothly merges into the surrounding Cartesian background mesh, which helps to avoid interpolation between the background and overlapping meshes. This work presents the results of testing this method, which showed its high accuracy. The disadvantages of the developed method include the limited types of fractured clusters for which this method can be applied since the overlapping meshes should not intersect. However, clusters of subvertical fractures are usually found in nature; therefore, the developed method is applicable in most cases.

**Keywords:** grid-characteristic method; overset grids; curvilinear meshes; fractures; geological faults; seismic prospecting; elastic waves; wave phenomena; waves scattering



## 1. Introduction

A large number of mineral deposits is associated with the presence of fluid-saturated fractures. This is often associated with volcanic zones [1]. Fluid-saturated fractured inclusions can be used to analyze mineral deposits [2]. Interpretation of seismic data obtained from mineral deposits requires the development of sufficiently effective methods to account for micro and macro fractures. Fault zones and fractured zones are quite common in the vicinity of open deposits. Seismic survey methods are used to explore ore deposits [3,4], so developing more accurate methods for seismic modeling and constructing mathematical models of geological environments is important for this area. However, multiple solutions to the corresponding direct problems are required to perform a full-wave inversion. This makes it necessary for scientists to develop computational methods for solving direct problems of the elastic wave equation that can describe the real geometry of a fractured cluster and conserve computing resources at the same time.

The accuracy of modeling the scattering of elastic waves on a fracture certainly depends on the mathematical model used. The Schoenberg linear slip model [5] (LSM) has proven to be consistent with practice [6,7] and is the most commonly used. We have also used it in this work. An anisotropic model [8] of an elastic medium is usually used and is acceptable at sufficiently long wavelengths. This mathematical model makes it possible

to reduce the number of computational resources, but not all types of waves scattered by fractures in nature are taken into account. Using the explicit method [9,10] is more accurate, and we use it in this work.

Fracture models differ in the computational grids used to describe the fracture itself. It is possible to distinguish such approaches as the use of Cartesian grids inside the fracture, codirectional with the coordinate axes [11,12], Cartesian grids and the model of infinitely thin fractures, codirectional [13–16] and not codirectional with the coordinate axes [17]. The approach proposed in [17] requires small coordinate steps to describe the fracture shape and correctly calculate the scattered waves. It can be concluded that the use of a Cartesian grid imposes restrictions on the orientation of fractures, which, in turn, will not make it possible to determine the important parameters of a fractured cluster, such as fracture orientation, fracture orientation spread, etc., or requires significant discretization to achieve sufficient calculation accuracy.

The methods for modeling incident and scattered elastic waves are divided into finite-difference methods [13,18–21] (which include, for example, the grid-characteristic method [12,16,17,22,23], and staggered-grid method [24–27]), finite element methods [28–30] (which include, for example, the discontinuous Galerkin method [31,32] and the method of spectral elements [33–36]). In addition, other methods, e.g., discrete particle schemes might be used [37].

Moreover, computational methods for modeling wave propagation in fractured geological media differ in the types of computational meshes used. These meshes can be Cartesian [13,17,23], curvilinear [28,37,38], unstructured triangular [9,23,28,39–42], tetrahedral [10,23,43], and quadrangular [40]. It can be concluded that only the use of Cartesian grids provides sufficient computational speed for solving real inverse problems of geophysics [44].

In [22], we proposed a grid-characteristic method on Chimera meshes. Chimera meshes [45,46] (also known as adaptive [47], overset [48], composite overlapping [49,50] meshes, or chimera method [51]) were used for the first time to solve problems of hydrodynamics [52–55]. Currently, they are used for other problems, for example, to solve the Poisson problem [56]. When using chimera meshes, the advantage is the Jacobian unit of the coordinate transformation in the grid surrounding the fracture due to the chimera mesh is Cartesian. Therefore, the method ensures the time step does not decrease. The disadvantage here is the need to interpolate the solution from the grid surrounding the fracture into the background computational grid. When using overlapping curvilinear meshes, which we propose in this work, the advantage is the absence of interpolation since the outside nodes of the curvilinear mesh coincide with the nodes of the background mesh, and the disadvantage is not the unit Jacobian of the coordinate transformation. In this work, we carry out the same tests as in work [22] to compare both approaches.

Curvilinear computational grids have been used for a long time in various fields. Let us consider applying computational curvilinear grids to the solution of elastic and acoustic wave equations in recent years. The method of pseudo-spectral elements to calculate the SH waves in a 2D case was used in [36]. The structured curvilinear meshes, also known as body-fitted and boundary-conforming meshes, were used to take into account the Earth's topography with the help of finite differences on modified staggered grids [27]. An efficient and accurate numerical algorithm for the simulation of borehole acoustic experiments using curvilinear meshes is presented in [38]. This type of grid is also used for accurate modeling of a free boundary of complex shapes [57].

This paper is organized as follows. The mathematical formulation of the problem and the features of the proposed modification of the numerical method are presented in Section 2. The features of the used overlapping curvilinear grids are discussed in Section 3. Section 4 discusses the testing of the proposed numerical method, and Section 5 presents the results of calculations of elastic wave scattering on geological models of fractured media of various complexity. The conclusions are presented in Section 6.

## 2. Computational Method

This section is devoted to the solved system of equations and the description of the proposed modification of the computational method. The features of constructing curvilinear computational grids surrounding the fractures in the geological media are considered in the next section.

### 2.1. Mathematical Statement

The conversion from the LMS model [5] of fractured media to the following initial-boundary value problem for the elastic wave equation is considered in detail in our previous work [22].

$$\rho \frac{\partial \mathbf{v}(\mathbf{r}, t)}{\partial t} = (\nabla \cdot \boldsymbol{\sigma}(\mathbf{r}, t))^{\mathrm{T}} \tag{1}$$

$$\frac{\partial \boldsymbol{\sigma}(\mathbf{r}, t)}{\partial t} = \left(\rho c_{\mathrm{P}}^2 - 2\rho c_{\mathrm{S}}^2\right)(\nabla \cdot \mathbf{v}(\mathbf{r}, t))\mathbf{I} + \rho c_{\mathrm{S}}^2 \left(\nabla \otimes \mathbf{v}(\mathbf{r}, t) + (\nabla \otimes \mathbf{v}(\mathbf{r}, t))^{\mathrm{T}}\right) \tag{2}$$

$$\boldsymbol{\sigma}^{\mathrm{L}}\left(\tilde{\mathbf{r}}, t\right) \cdot \mathbf{m}\left(\tilde{\mathbf{r}}\right) = \boldsymbol{\sigma}^{\mathrm{R}}\left(\tilde{\mathbf{r}}, t\right) \cdot \mathbf{m}\left(\tilde{\mathbf{r}}\right), \tag{3}$$

$$\boldsymbol{\sigma}^{\mathrm{R}}\left(\tilde{\mathbf{r}}, t\right) \cdot \mathbf{m}\left(\tilde{\mathbf{r}}\right) - \left(\mathbf{m}\left(\tilde{\mathbf{r}}\right) \cdot \boldsymbol{\sigma}^{\mathrm{R}}\left(\tilde{\mathbf{r}}, t\right) \cdot \mathbf{m}\left(\tilde{\mathbf{r}}\right)\right) = 0, \tag{4}$$

$$\mathbf{v}^{\mathrm{L}}\left(\tilde{\mathbf{r}}, t\right) \cdot \mathbf{m}\left(\tilde{\mathbf{r}}\right) = \mathbf{v}^{\mathrm{R}}\left(\tilde{\mathbf{r}}, t\right) \cdot \mathbf{m}\left(\tilde{\mathbf{r}}\right). \tag{5}$$

Here and further in the text, $\mathbf{v}(\mathbf{r}, t)$, $\boldsymbol{\sigma}(\mathbf{r}, t)$ represent the unknowns of the velocity and symmetric Cauchy stress tensor of the second rank, $\rho$ denotes the density, $c_{\mathrm{P}}$, $c_{\mathrm{S}}$ represent the speeds of the pressure (P-) and shear (S-) elastic waves, respectively; $\mathbf{I}$ means the unit tensor of the second rank; $\otimes$ denotes the tensor product of vectors, $(\mathbf{a} \otimes \mathbf{b})_{ij} = a_i b_j$; $\mathbf{r}$, $t$ are the coordinate vector in the integration domain and the time, respectively; $\tilde{\mathbf{r}} \in \Gamma_n, n = 1, N$ denotes coordinates of infinity thin fracture $\Gamma_n$; and $\mathbf{m}\left(\tilde{\mathbf{r}}\right)$ denotes the unit vector normal to the fracture $\Gamma_n$ in the point $\tilde{\mathbf{r}}$.

Note that when using this type of notation, the presented initial-boundary value problem has the same form in both three-dimensional and two-dimensional cases.

Different types of sources were used, e.g., a plane P-wave as an initial condition or point sources and zero initial conditions. The source type is described in Section 5 for each specific problem being solved.

### 2.2. Grid-Characteristic Method on Structured Curvilinear Meshes

We introduce the coordinate transformation $(\xi_1(x, y), \xi_2(x, y))$. The structured curvilinear grid in the introduced coordinate system will turn into a Cartesian grid with a unit step in coordinates. Herewith, we know the positions of the nodes of the structured curvilinear grid in the original coordinate system.

$$(x_n, y_m), n \in [1, N], m \in [1, M]. \tag{6}$$

There is no need to look for a coordinated transformation in the analytical form. We numerically calculate all quantities necessary for subsequent calculations. That is, for each node of the structured curvilinear grid and each direction $j$, we introduce the following vector $\mathbf{n}_{n,m}^j$:

$$\mathbf{n}_{n,m}^j = \mathbf{n}^j(x_n, y_m) = \frac{\nabla \xi_j(x_n, y_m)}{\left|\nabla \xi_j(x_n, y_m)\right|} = \frac{\nabla \xi_j(x_n, y_m)}{l_{n,m}^j}. \tag{7}$$

$$\nabla \xi_j(x_n, y_m) \approx \begin{cases} \left[\frac{x_{n+1}-x_{n-1}}{2}, \frac{y_{m+1}-y_{m-1}}{2}\right]^{\mathrm{T}}, n \in [2, N-1], m \in [2, M-1] \\ \left[x_{n+1}-x_n, \frac{y_{m+1}-y_{m-1}}{2}\right]^{\mathrm{T}}, n = 1, m \in [2, M-1] \\ \left[x_n - x_{n-1}, \frac{y_{m+1}-y_{m-1}}{2}\right]^{\mathrm{T}}, n = N, m \in [2, M-1] \\ \left[\frac{x_{n+1}-x_{n-1}}{2}, y_{m+1}-y_m\right]^{\mathrm{T}}, n \in [2, N-1], m = 1 \\ \left[\frac{x_{n+1}-x_{n-1}}{2}, y_m - y_{m-1}\right]^{\mathrm{T}}, n \in [2, N-1], m = M \end{cases}. \tag{8}$$

Furthermore, we introduce vectors $\mathbf{n}_{n,m}^{1,j}$ as being perpendicular to the vectors $\mathbf{n}_{n,m}^{j}$. Now, we can consider a set of symmetric tensors of the second rank for each pair of vectors $\mathbf{n}_{n,m}^{j}$ and $\mathbf{n}_{n,m}^{1,j}$:

$$\mathbf{N}_{n,m}^{\alpha,\beta,j}(x, y) = \frac{1}{2}\left(\mathbf{n}_{n,m}^{\alpha,j} \otimes \mathbf{n}_{n,m}^{\beta,j} + \mathbf{n}_{n,m}^{\beta,j} \otimes \mathbf{n}_{n,m}^{\alpha,j}\right), \alpha = 0, 1, \beta = 0, 1. \tag{9}$$

Here, $\mathbf{n}_{n,m}^{0,j} \equiv \mathbf{n}_{n,m}^{j}$.

Further, for each point of the curvilinear grid, the formulae from our previous work [22] can be applied to calculate unknown values for each of the directions.

### 2.3. Computational Algorithm

In the proposed modification of the grid-characteristic method using overlapping curvilinear grids, we use the following computational algorithm for every time step $n$.

1. Values from the background grid are copied in two layers of nodes placed along the vertical boundaries of the curvilinear grids and surrounding fractures. Section 4 describes in detail the features of constructing these curvilinear grids in the proposed modification of the computational method, which makes it possible to carry out this copying and, at the same time, obtain high accuracy of calculations.
2. Calculations are carried out in the direction OX in the background grid by the grid-characteristic method on Cartesian grids, described in [22].
3. Calculations are also performed for the OX direction in each curvilinear meshes surrounding the fractures using the grid-characteristic method on structured curvilinear meshes described in the next section.
4. Values from two layers of nodes of curvilinear grids placed along the vertical boundaries are copied in the congruent position nodes of the background grid.

Then, all these four steps are performed for the direction OY and the horizontal boundaries.

This algorithm can be easily generalized to the three-dimensional case. The boundaries will be replaced herewith by the corresponding planes perpendicular to the directions under consideration.

Below we provide a pseudocode showing the sequence of implementation of the various stages of the algorithm in the software. When implemented, it is necessary to carry out the sequence of interpolation correctly to the boundary nodes of overlapping curvilinear meshes with fractures and to the nodes of the background mesh from the meshes with fractures within the time integration cycle. The sequence is displayed in pseudocode.

## 3. Features of Curvilinear Computational Meshes

A curvilinear computational mesh is built from the following principles. The outer two rows of curved mesh cells strictly coincide with the background mesh cells to not use interpolation but copy from these two rows to the background mesh (Figure 1A, red (black) nodes) and back to the ghost nodes of the curvilinear mesh from the background mesh (Figure 1A, blue (grey) nodes). Next, a third-order polynomial is constructed to ensure the continuity of the first derivative on the line marked in color and thickness in Figure 1B. The areas of the third-order polynomial are marked in red lines of middle thickness, and the

crack area is shown in green lines with maximal thickness in the center. The blue lines with maximal thickness near the edges ensure an exact match with the background grid.

| Pseudocode |
| --- |

1. Loading grids and computing the Jacobian of coordinate transformations;
2. Preparing data for interpolation;
3. Loading of geological model and calculation data.
4. Time Integration Cycle

    4.1. Interpolation of data into ghost nodes of grids with fractures;
    4.2. Calculation of data into nodes of background grid;
    4.3. Calculation of data into nodes of grids with fractures;
    4.4. Interpolation of data into nodes of background grid from data into nodes of grids with fractures;
    4.5. Boundary correctors;
    4.6. Saving the result at the current time moment.

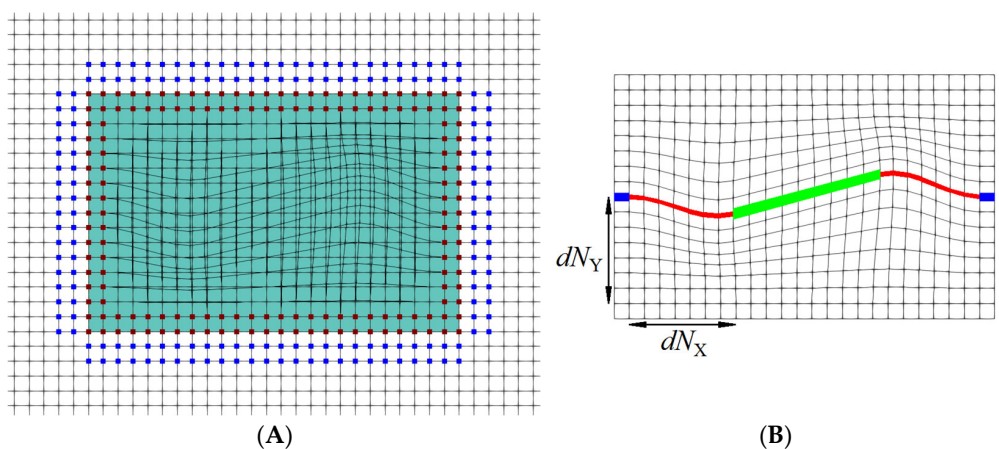

(**A**)            (**B**)

**Figure 1.** (**A**) Scheme of copying: from the red nodes of the curvilinear mesh to the background mesh and from the blue nodes of the background mesh to the ghost nodes of the curvilinear mesh; (**B**) curvilinear mesh around the fracture.

Next, for the sake of simplicity, consider the mesh for a fracture with an angle less than 45° the horizontal (Figure 1B). The number of cells around the crack is chosen automatically so that the vertical distance $dN_Y$ between the cells does not exceed a certain $h_{MIN}$ (Figures 1B and 2A). The number of nodes along the horizontal $dN_X$ is chosen so that the inclination angle between the adjacent cells along the vertical edge does not exceed a fixed angle β (Figures 1B and 3). The vertical step in the grid changes smoothly according to the following function:

$$f(x) = \begin{cases} f(-x), \ x < 0 \\ 1 - 2x^2, \ 0 \le x < 0.5 \\ 2(x-1)^2, \ 0.5 \le x < 1 \\ 0, \ x \ge 1 \end{cases}$$

(10)

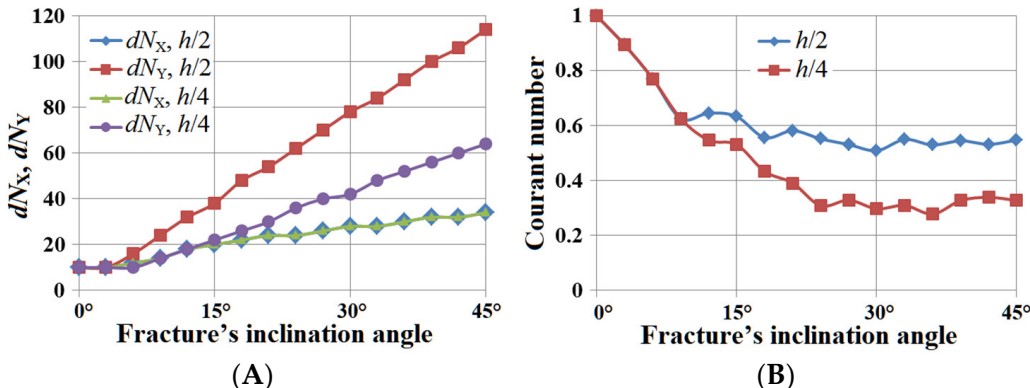

**Figure 2.** (**A**) Graphs of $dN_X$ and $dN_Y$ on the fracture's inclination angle for $h_{MIN} = h/2$ and $h/4$, $\beta = 15°$; (**B**) graphs of the Courant number in background mesh on the fracture's inclination angle for $h_{MIN} = h/2$ and $h/4$, $\beta = 15°$.

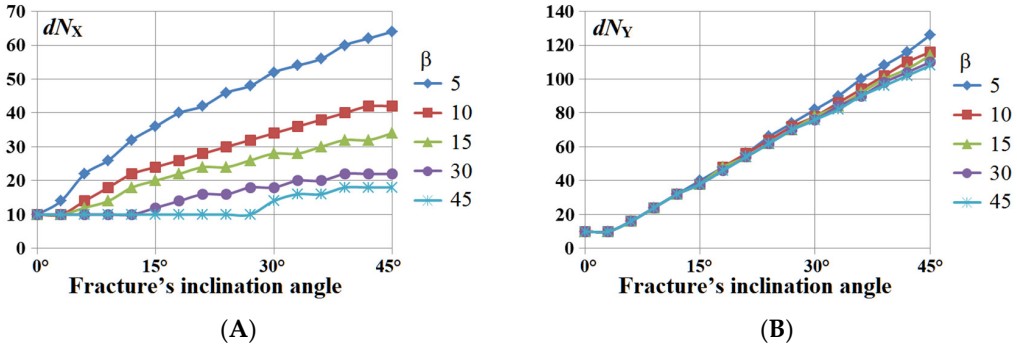

**Figure 3.** (**A**) Graphs of $dN_X$ on the fracture's inclination angle $\beta = 5°, 10°, 15°, 30°, 45°$, $h_{MIN} = h/2$; (**B**) graphs of $dN_Y$ on the fracture's inclination angle for $\beta = 5°, 10°, 15°, 30°, 45°$, $h_{MIN} = h/2$.

Note that in Figures 2A and 3 minimum $dN_X$ and $dN_Y$ values are 10.

Figure 2A demonstrates at what angles of inclination of the fracture it is suitable to use $h_{MIN} = h/2$, and at what angles it is suitable to use $h_{MIN} = h/4$. The Courant number in the background mesh required for the stability of the method on a curvilinear mesh is close to the ratio $h_{MIN}/h$ (Figure 2B) for $\beta = 15°$, and $\beta = 15°$ is the optimal parameter of the curvilinear mesh construction. The suitability of $\beta = 15°$ is also confirmed by the graphs in Figure 3 since, for this value the size of the curvilinear mesh is also optimal.

Let us provide some examples of computational structured curvilinear meshes around the fractures for the different fracture's inclination angles (Figure 4). These fractures are colored red. There are 10 nodes of the Cartesian background grid per the fracture for all inclination angles. One can see the dependence of the number of nodes on the fracture's inclination angle.

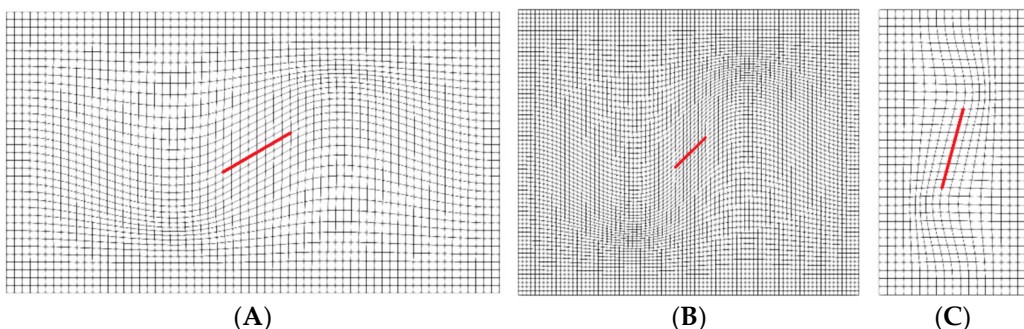

**Figure 4.** Examples of structured curvilinear meshes around fractures for different fracture inclination angles: (**A**) 30°; (**B**) 45°; (**C**) 75°.

## 4. Testing

In this section, we study the quality of calculation of the types of waves reflected from a crack depending on the angle between the crack and the OX axis. To do this, we compare synthetic seismograms obtained from receivers located on a circle around the center of the fracture (Figure 5A), calculated using the proposed modification of the grid-characteristic method on overlapping curvilinear meshes and calculated using the grid-characteristic method on Cartesian grids. The Ricker wavelet point source with a frequency of 20 Hz was placed near the first receiver (Figure 5A). To model the fracture's inclination and use a fracture co-directed to the OX axis, we rotate the circles on which the source and receiver lie in accordance with Figure 5B. The angle $\alpha$ is the angle between the fracture and the horizontal axis OX for Figure 5A and the inclination angle for Figure 5B, respectively.

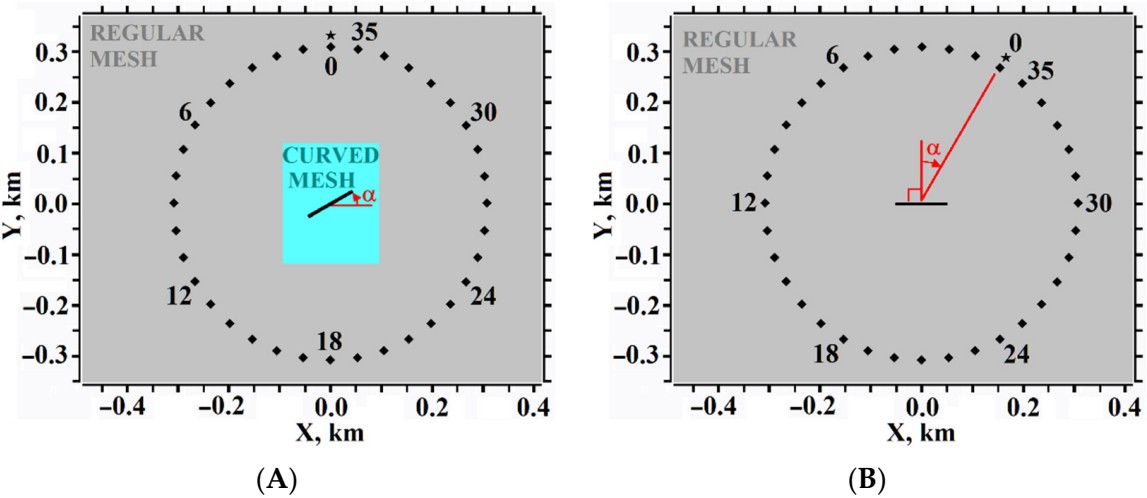

(A)　　　　　　　　　　　　　　　　　　　　　(B)

**Figure 5.** Two interconvertible problem statements, the receivers' positions are marked with the diamonds, and the source position is marked with the asterisk: (**A**) with an overlapping curvilinear mesh, original coordinate system; (**B**) without an overlapping curvilinear mesh.

We used the coordinate step of 2 m, the fracture length of 100 m, the P-wave speed of 1000 m/s, the S-wave speed of 600 m/s, and the density of 1000 kg/m$^3$. The maximum time step in all calculations was equal to 0.8 ms. Note that the time step depends on the angle $\alpha$ due to stability conditions. This issue was discussed in Section 3.

The dependence of the relative errors on $\alpha$ is shown in Figure 6. We have used the following formulae to calculate relative errors:

$$
\mathrm{E_V\{L_1\}} = \frac{\sum_{i=1}^{N_R}\sum_{j=1}^{N_T}\left(\left(v_X^{i,j} - V_X^{i,j}\right)^2 + \left(v_Y^{i,j} - V_Y^{i,j}\right)^2\right)^{\frac{1}{2}}}{\sum_{i=1}^{N_R}\sum_{j=1}^{N_T}\left(\left(V_X^{i,j}\right)^2 + \left(V_Y^{i,j}\right)^2\right)^{\frac{1}{2}}}, \tag{11}
$$

$$
\mathrm{E_V\{L_\infty\}} = \frac{\max\limits_{i\in[1,N_R],j\in[1,N_T]}\left(\left(v_X^{i,j} - V_X^{i,j}\right)^2 + \left(v_Y^{i,j} - V_Y^{i,j}\right)^2\right)^{\frac{1}{2}}}{\max\limits_{i\in[1,N_R],j\in[1,N_T]}\left(\left(V_X^{i,j}\right)^2 + \left(V_Y^{i,j}\right)^2\right)^{\frac{1}{2}}}. \tag{12}
$$

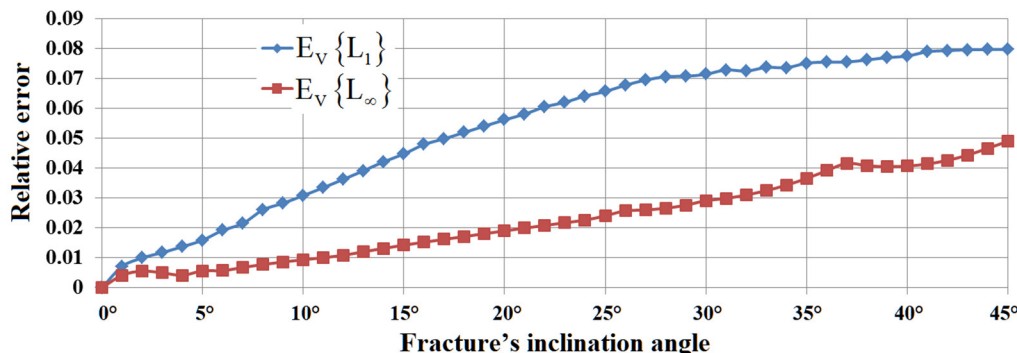

**Figure 6.** Dependence of the relative errors on the angle $\alpha$.

In (11) and (12), $v_X^{i,j}$, $v_Y^{i,j}$ denote the velocity field components obtained using the proposed method in the original coordinate system; $V_X^{i,j}$, $V_Y^{i,j}$ represent the velocity field components obtained without an overlapping curvilinear mesh in the coordinate system rotated on angle $\alpha$. In (11) and (12) and further in the text, $N_R$ denotes the number of receivers that equals 36; $N_T$ denotes the number of time steps that depends on the time step in considered calculation with considered $\alpha$. $N_T$ depends on $\alpha$ due to the constant total time was used for each $\alpha$, and different time steps were used.

The dependence of the relative errors of the anomalous velocity on the angle $\alpha$ is shown in Figure 7. To calculate the relative errors of the anomalous field, we used the following formulae:

$$E_A\{L_1\} = \frac{\sum\limits_{i=1}^{N_R}\sum\limits_{j=1}^{N_T}\left(\left(a_X^{i,j} - A_X^{i,j}\right)^2 + \left(a_Y^{i,j} - A_Y^{i,j}\right)^2\right)^{\frac{1}{2}}}{\sum\limits_{i=1}^{N_R}\sum\limits_{j=1}^{N_T}\left(\left(A_X^{i,j}\right)^2 + \left(A_Y^{i,j}\right)^2\right)^{\frac{1}{2}}}, \tag{13}$$

$$E_A\{L_\infty\} = \frac{\max\limits_{i\in[1,N_R],j\in[1,N_T]}\left(\left(a_X^{i,j} - A_X^{i,j}\right)^2 + \left(a_Y^{i,j} - A_Y^{i,j}\right)^2\right)^{\frac{1}{2}}}{\max\limits_{i\in[1,N_R],j\in[1,N_T]}\left(\left(A_X^{i,j}\right)^2 + \left(A_Y^{i,j}\right)^2\right)^{\frac{1}{2}}}. \tag{14}$$

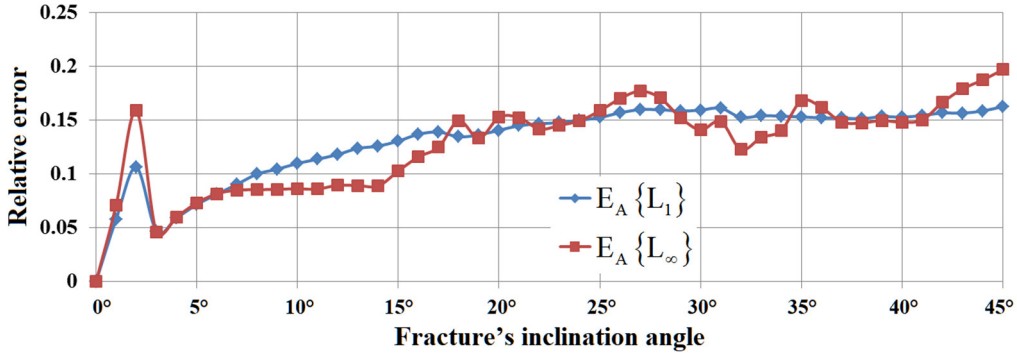

**Figure 7.** Dependence of the relative errors of the anomalous velocity on the angle $\alpha$.

In (13) and (14), $a_X^{i,j}$, $a_Y^{i,j}$ are the anomalous velocity field components calculated using the proposed method of the original coordinate system; $A_X^{i,j}$, $A_Y^{i,j}$ are the anomalous velocity field components of the calculated without an overlapping curvilinear mesh in the respectively rotated coordinate system.

The examples of velocity field snapshots and seismograms for the angle α of 30° are shown in Figures 8–10.

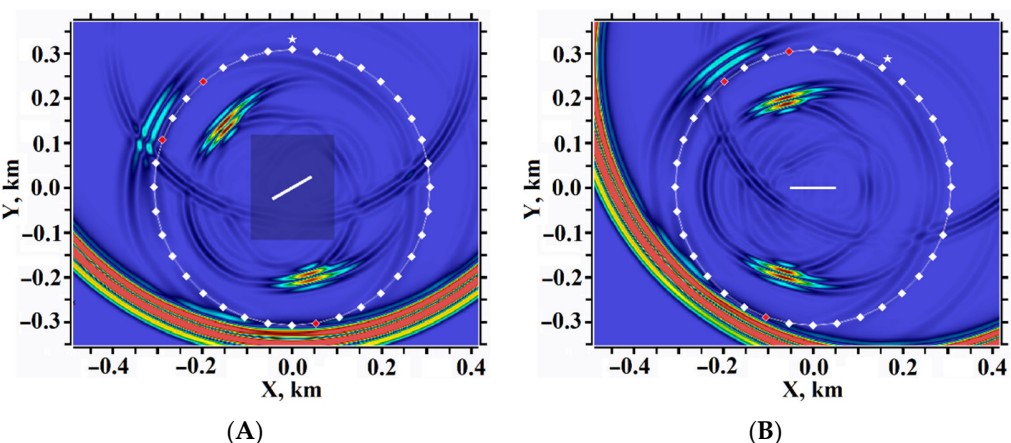

(A)  (B)

**Figure 8.** Velocity modulus snapshots, time moment of 0.8 s, distance in km, diamonds mark the receivers' positions, the asterisk denotes the source position: (**A**) with an overlapping curvilinear mesh;(**B**) without an overlapping curvilinear mesh.

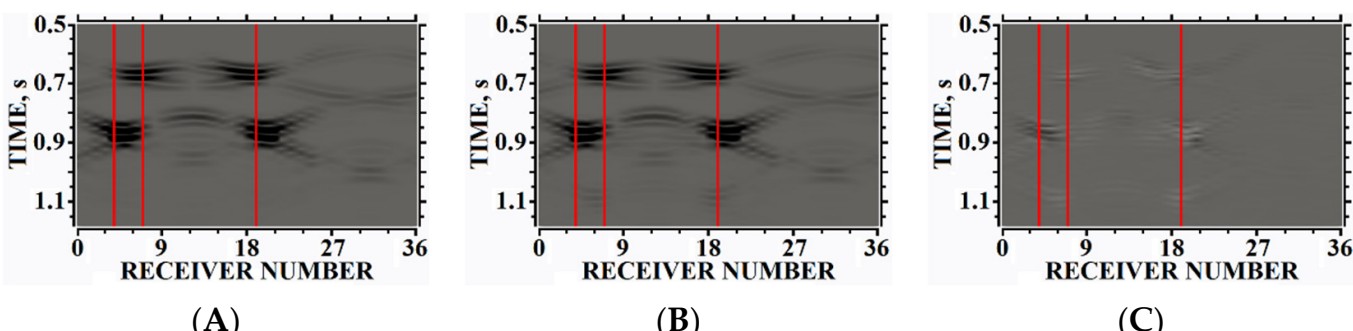

(A)  (B)  (C)

**Figure 9.** Seismograms, anomalous velocity modulus: (**A**) with an overlapping curvilinear mesh; (**B**) without an overlapping curvilinear mesh; (**C**) difference.

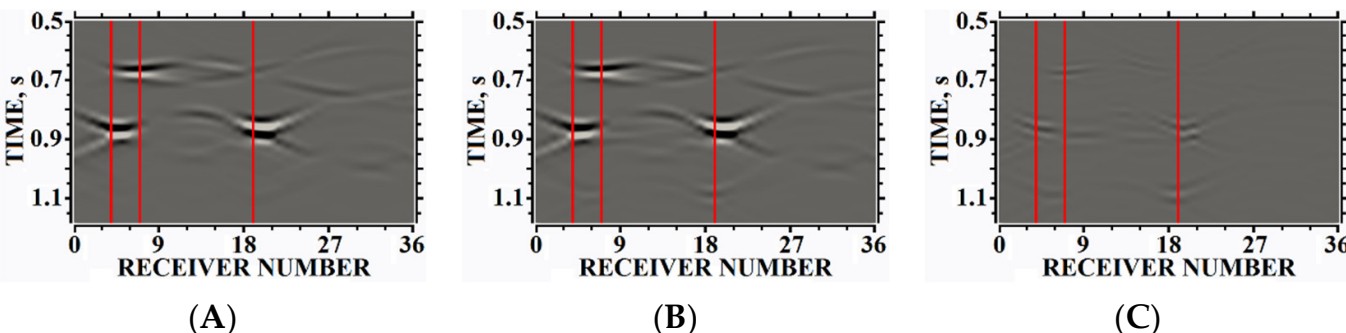

(A)  (B)  (C)

**Figure 10.** Seismograms, X-component of the anomalous velocity: (**A**) with an overlapping curvilinear mesh; (**B**) without an overlapping curvilinear mesh; (**C**) difference.

The testing performed demonstrates the high accuracy of the proposed numerical method.

We specifically used such a discretization from 17 to 25 points (in dependence on the inclination between the wavefront and the coordinate axis) of the background computational mesh are used to treat the P-waves, and from 10 to 15 points are used to treat the S-waves. This allows for a more representative comparison of the proposed method with available alternatives, including the one presented in [22].

To clarify how specifically Figures 8–11 differ, we present a series of one-dimensional plots in Figures 12–20.

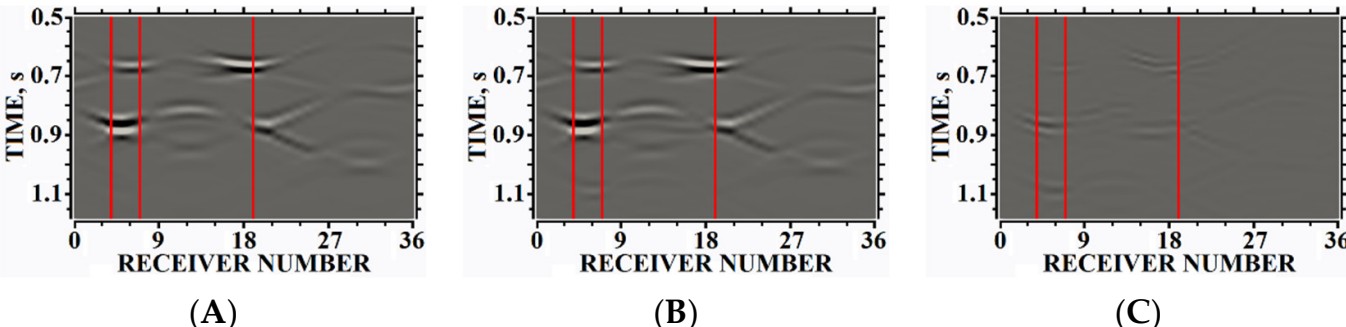

**Figure 11.** Seismograms, Y-component of the anomalous velocity: (**A**) with overlapping curvilinear mesh; (**B**) without overlapping curvilinear mesh; (**C**) difference.

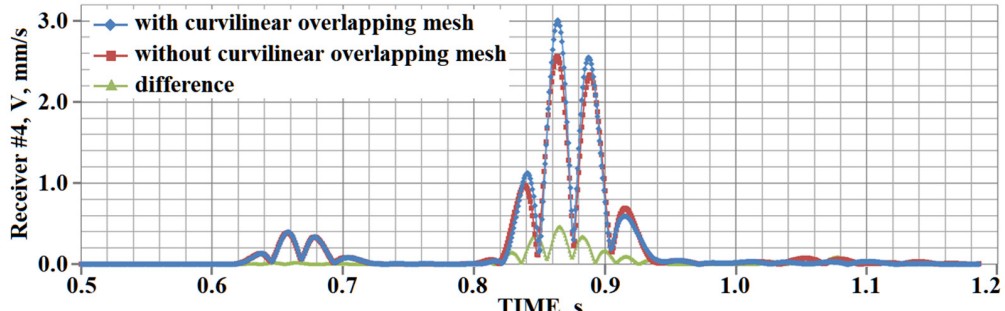

**Figure 12.** Receiver #4, dependencies of velocity modulus on time.

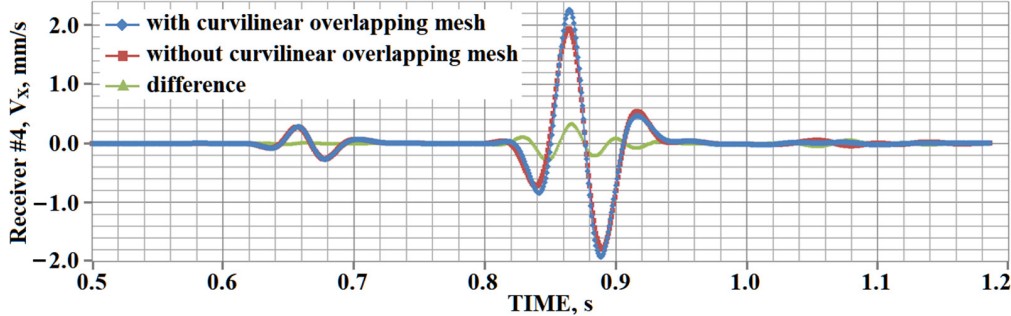

**Figure 13.** Receiver #4, dependencies of X-component of velocity on time.

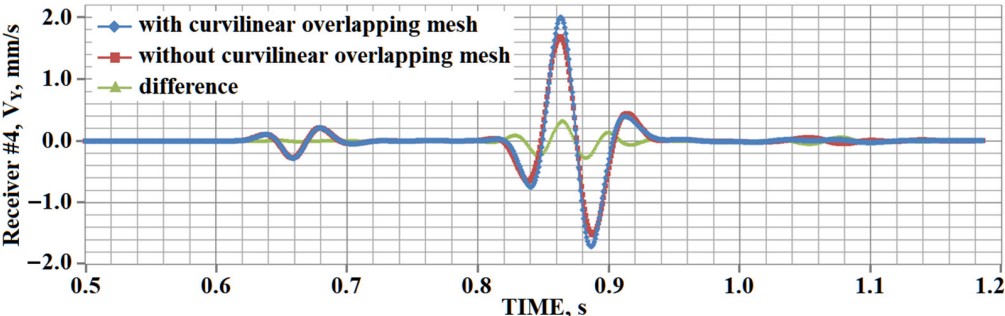

**Figure 14.** Receiver #4, dependencies of Y-component of velocity on time.

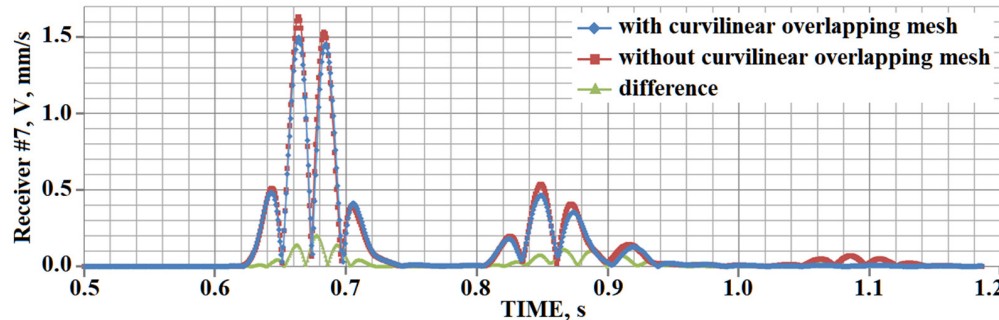

**Figure 15.** Receiver #7, dependencies of velocity modulus on time.

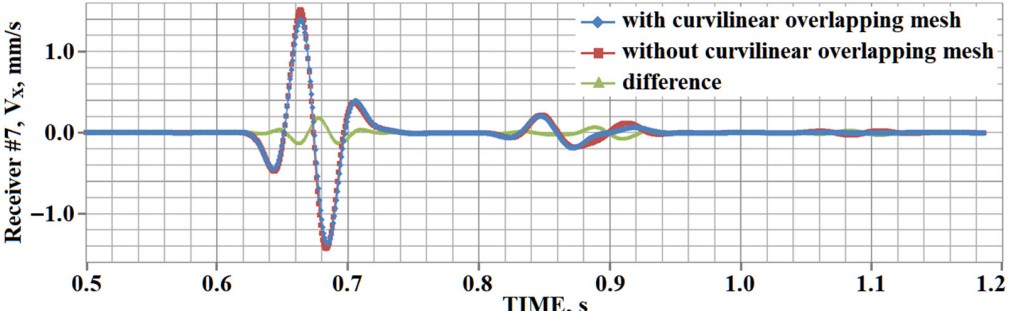

**Figure 16.** Receiver #7, dependencies of X-component of velocity on time.

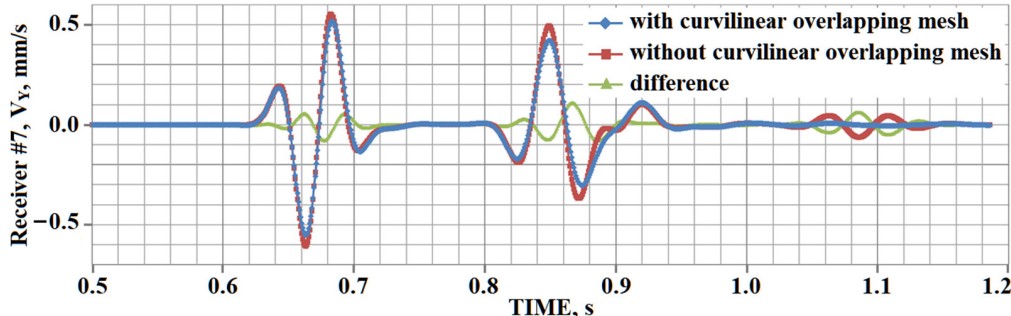

**Figure 17.** Receiver #7, dependencies of Y-component of velocity on time.

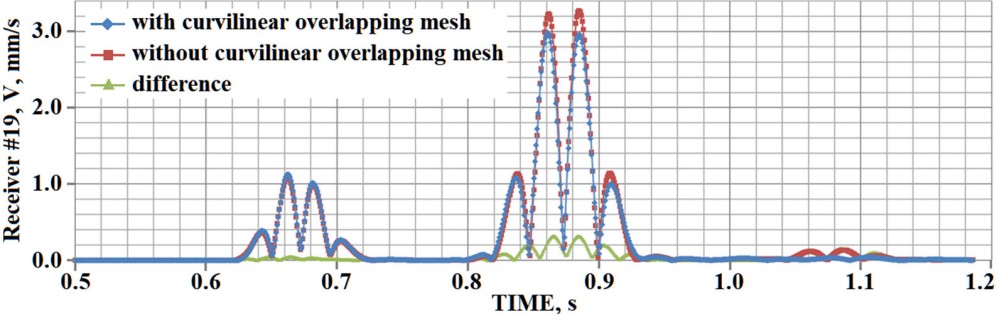

**Figure 18.** Receiver #19, dependencies of velocity modulus on time.

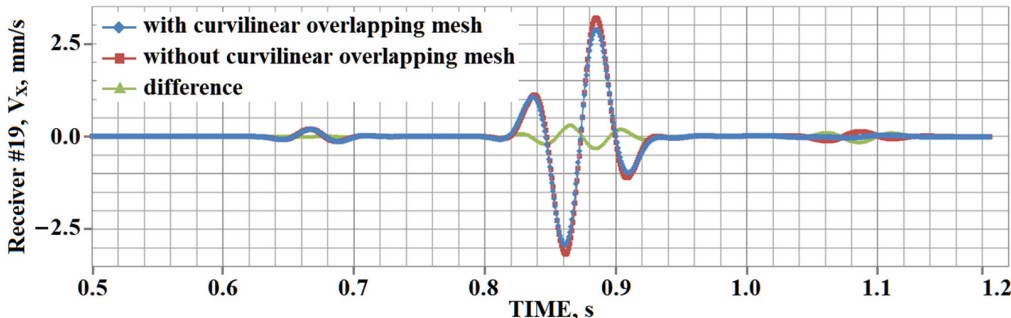

**Figure 19.** Receiver #19, dependencies of X-component of velocity on time.

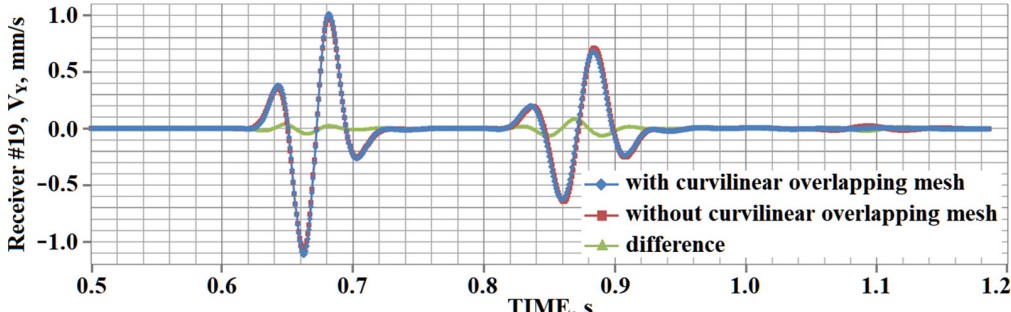

**Figure 20.** Receiver #19, dependencies of Y-component of velocity on time.

The data on the plots in Figures 12–20 are registered by the receivers marked by red color in Figure 8 and correspond to vertical lines on the seismograms in Figures 9–11.

By the X- and Y-components of velocity, we mean the components in the non-rotated coordinate system introduced in Figure 5A. Thus, to obtain the plotted data, the original data for the problem «without overlapping curvilinear mesh» were transformed by rotating by the appropriate angle of 30°.

The choice of receiver numbers for plotting is due to the following factors. Figures 12–14 (Receiver #4), Figures 18–20 (Receiver #19) correspond to two converted PS waves, respectively. Figures 15–19 (Receiver #7) correspond to PP-wave reflected from the fracture.

## 5. Results of Numerical Experiments

In this section, we present the results of the test calculations for the different geological models of fractured media of varying complexity. We took similar geological models as in [22] to compare the results obtained.

### 5.1. Example #1

A set of fractures with arbitrary length, location, and orientation (Figures 21–24) is considered in this section. This model is based on the one given in [39]. The initial conditions of a plane elastic P-wave (Figures 21 and 22) or S-wave (Figures 23 and 24) with a wavelet length being equal to 50 m, half-sine wave profile (effective frequency of 50 Hz, real frequency of 25 Hz), and a unit amplitude was applied. We used a coordinate step of 1 m and a time step of 0.16 ms. In [22], we used a time step of 0.3 ms in a similar test. The P-wave speed of 3000 m/s, the S-wave speed of 2000 m/s, and the density of 2500 kg/m$^3$ were used in this section and Sections 5.2 and 5.3.

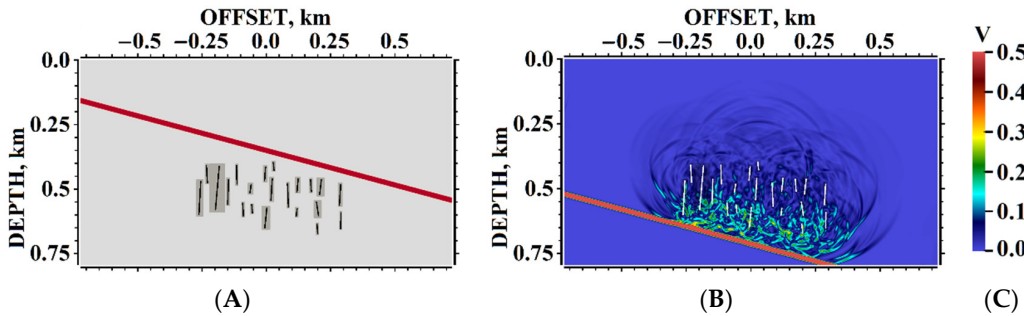

**Figure 21.** Example #1, incident P-wave: (**A**) geophysical model, overlapping curvilinear meshes, the incident P-wave is marked with the red line; (**B**) velocity modulus snapshot at the time moment 0.26 s; (**C**) scale of the velocity modulus used in the paper.

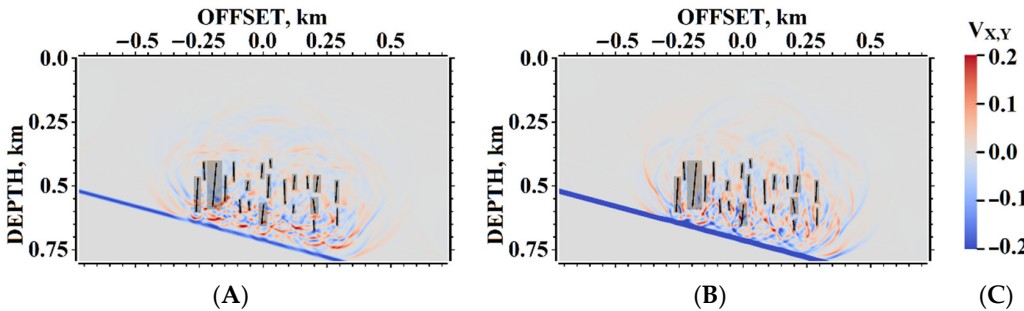

**Figure 22.** Example #1, incident P-wave, snapshots at the time moment 0.26 s: (**A**) horizontal component of velocity; (**B**) vertical component of velocity; (**C**) scale of the velocity components used in the paper.

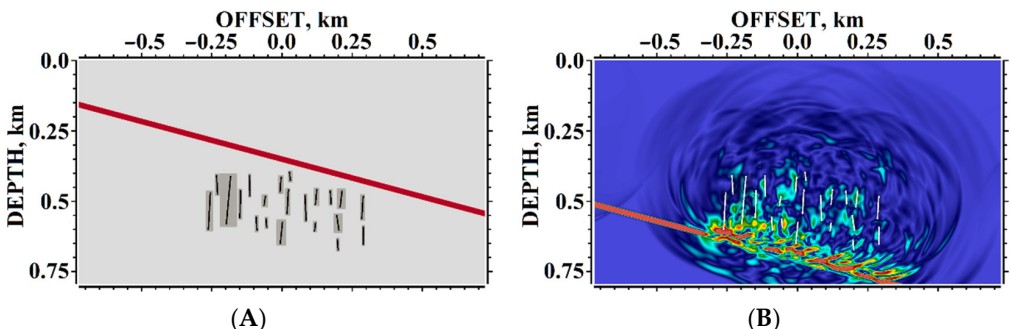

**Figure 23.** Example #1, incident S-wave: (**A**) geophysical model, overlapping curvilinear meshes, the incident S-wave is marked with the red line; (**B**) velocity modulus snapshot at the time moment 0.38 s.

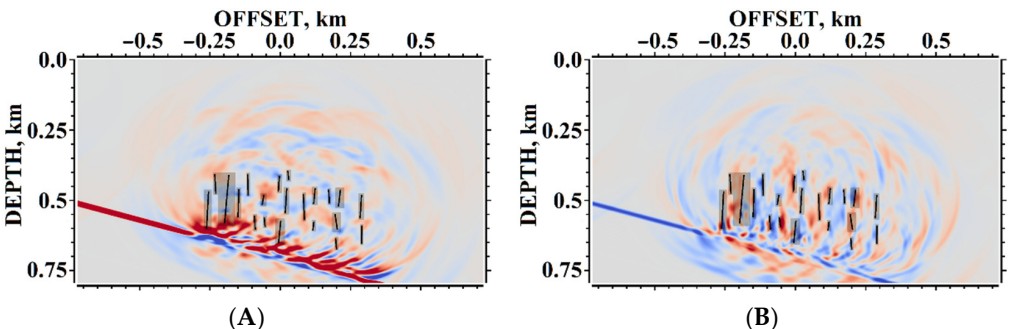

**Figure 24.** Example #1, incident S-wave, snapshots at the time moment 0.38 s: (**A**) horizontal component of velocity; (**B**) vertical component of velocity.

The presented results show a more significant dispersion of the transmitted S-waves compared to the P-waves. It can also be seen that the developed method managed well with modeling the passage of plane waves through a cluster of multidirectional fractures of different sizes. However, the model from [39] had to be modified so that the overlapping curvilinear meshes surrounding the fractures did not intersect with each other.

### 5.2. Example #2

The model from [9] is discussed in this section. We used a point source marked by the red dot in Figure 25A with a source function of Ricker wavelet; a peak frequency equals 30 Hz. The coordinate step of 2 m, and the time of 0.2 ms were used. In [22], in a similar test, we used a time step of 0.6 ms for the same coordinate step in the background mesh. In [9] in the similar test, the time step was equal to 1 ms for triangular mesh with an edge length varying around 4 m. Figures 25 and 26. show the used model and snapshots of the calculated elastic wave field at the time moment of 0.555 s.

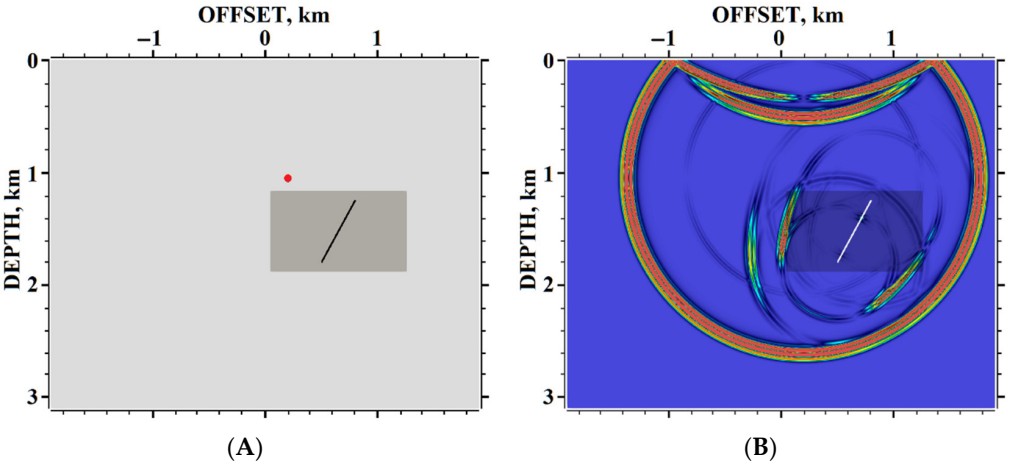

**Figure 25.** Example #2: (**A**) geophysical model, overlapping curvilinear mesh, the point source is marked by the red dot; (**B**) velocity modulus snapshot.

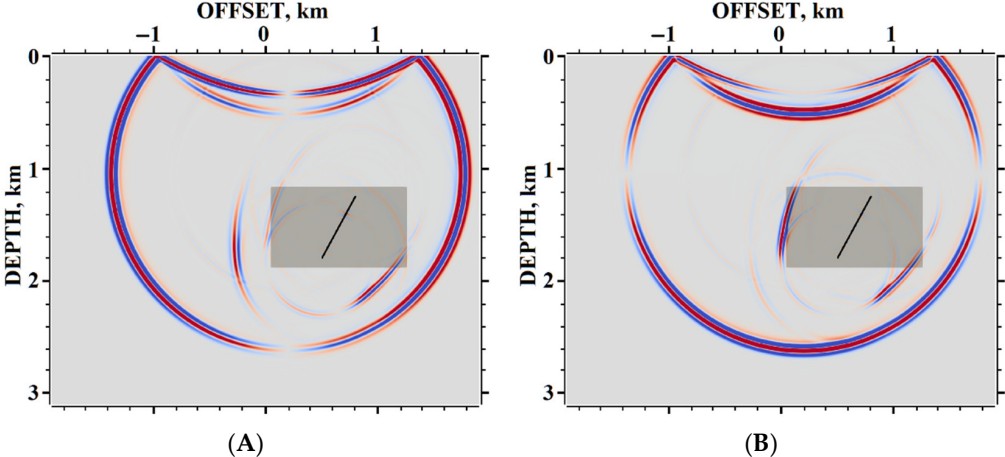

**Figure 26.** Example #2, snapshots at the time moment of 0.555 s: (**A**) horizontal component of velocity; (**B**) vertical component of velocity.

There is good qualitative conformity on all types of waves with the results of works [9,22]. One can also see all types of waves that should be observed with this scattering [58].

### 5.3. Example #3

This section discusses the model from [9]. We model wave propagation in the fractured zone, in which there are 101 fractures rotated at 30° with a horizontal distance of 42 m

between them. We used the same point source as in the previous Section 5.2, the coordinate step of 2 m, and the time step of 0.16 ms. In [22], in a similar test, we used a time step of 0.6 ms for the same coordinate step in the background mesh. In [9], in the similar test, the time step was equal to 1 ms for triangular mesh with an edge length varying around 4 m. Figures 27 and 28. show the used model and snapshots of the calculated elastic wave field snapshots at the time moment of 0.86 s.

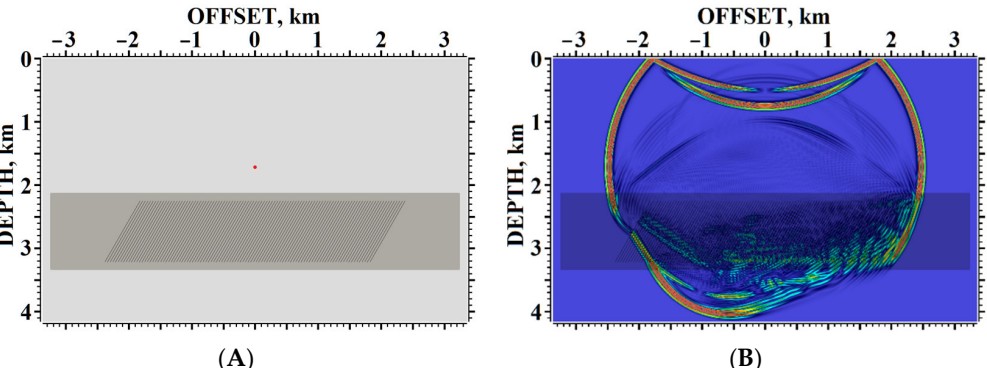

**Figure 27.** Example #3: (**A**) geophysical model, overlapping curvilinear mesh, a point source is marked by the red dot; (**B**) velocity modulus snapshot.

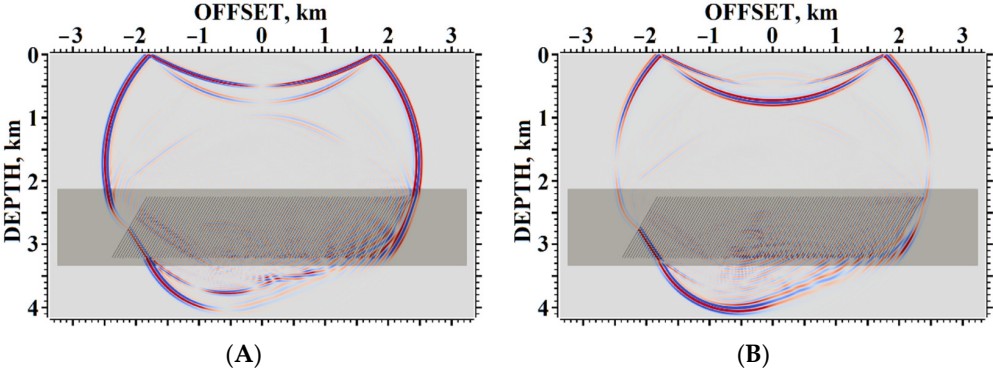

**Figure 28.** Example #3, snapshots at the time moment of 0.86 s: (**A**) horizontal component of velocity; (**B**) vertical component of velocity.

There is good qualitative conformity on all types of waves with the results of works [9,22] and the same observed significant dispersion of the incident P-wave on the fractured zone.

### 5.4. Example #4

In this section, we discuss the geological model based on one from [39]. In our model, the fractures are subvertical. We used the coordinate step of 0.78125 m, and the time step of 50 μs; they are the same as in [22]. Figures 29 and 30 show the model and the calculated elastic wave field snapshots at a time moment of 0.25 s.

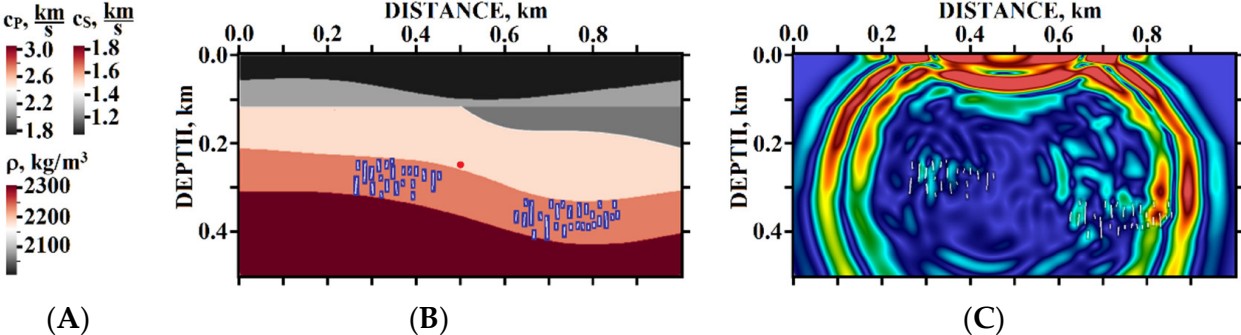

**Figure 29.** Example #4: (**A**) scales of elastic parameters; (**B**) geophysical model, overlapping curvilinear meshes, the point source is marked by the red dot; (**C**) velocity modulus snapshot.

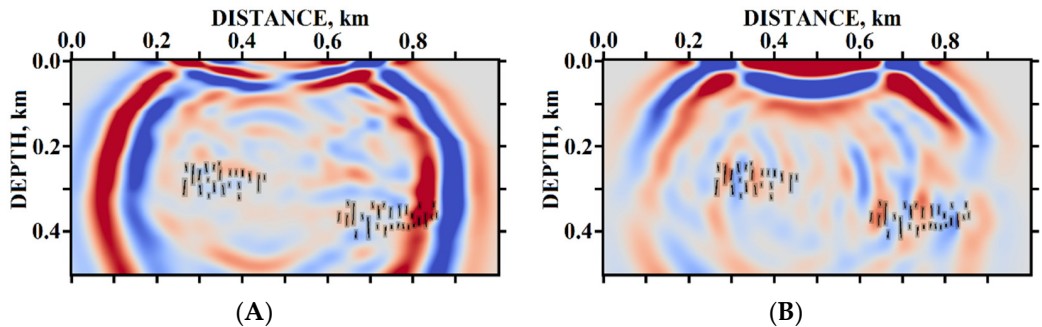

**Figure 30.** Example #4, snapshots at the time of 0.25 s: (**A**) horizontal component of velocity; (**B**) vertical component of velocity.

One can conclude that the proposed grid-characteristic method using overlapping curvilinear meshes is limited by the size of the curvilinear mesh surrounding the fracture and is well suited for describing only a certain type of fracture cluster, e.g., the clusters of subvertical fractures.

*5.5. Example #5*

In this section, we consider the model from [4]. The study aims to show how fractures in the geological media affect wave fields and seismograms arising from metal ore bodies seismic exploration. To do this, we placed two types of fractured zones above and next to the metal ore body. Accordingly, we obtain three calculations: «without fractured zone» (Figure 31A), the fractured zone above the metal ore body («fractured zone #1», Figure 32A), and the fractured zone next to the metal ore body («fractured zone #2», Figure 32B). Below, in Figures 33–35, snapshots of wave fields and seismograms are presented.

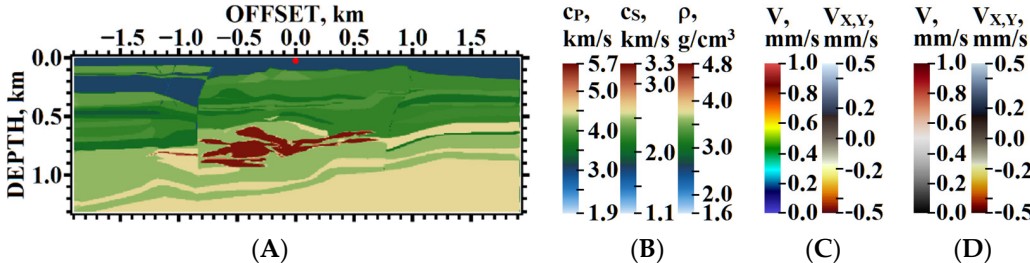

**Figure 31.** Example #5: (**A**) geophysical model, «without fractures», the point source is marked by the red dot; (**B**) scales of elastic properties; (**C**) scales of wave field snapshots; (**D**) scales of seismograms.

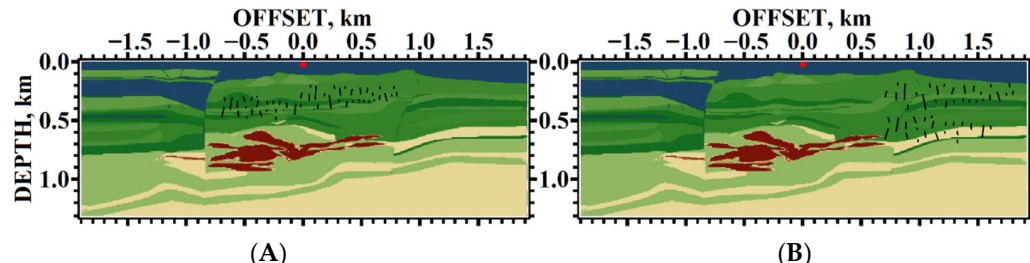

**Figure 32.** Example #5, geophysical model, the point source is marked by the red dot: (**A**) «fractured zone #1»; (**B**) «fractured zone #2».

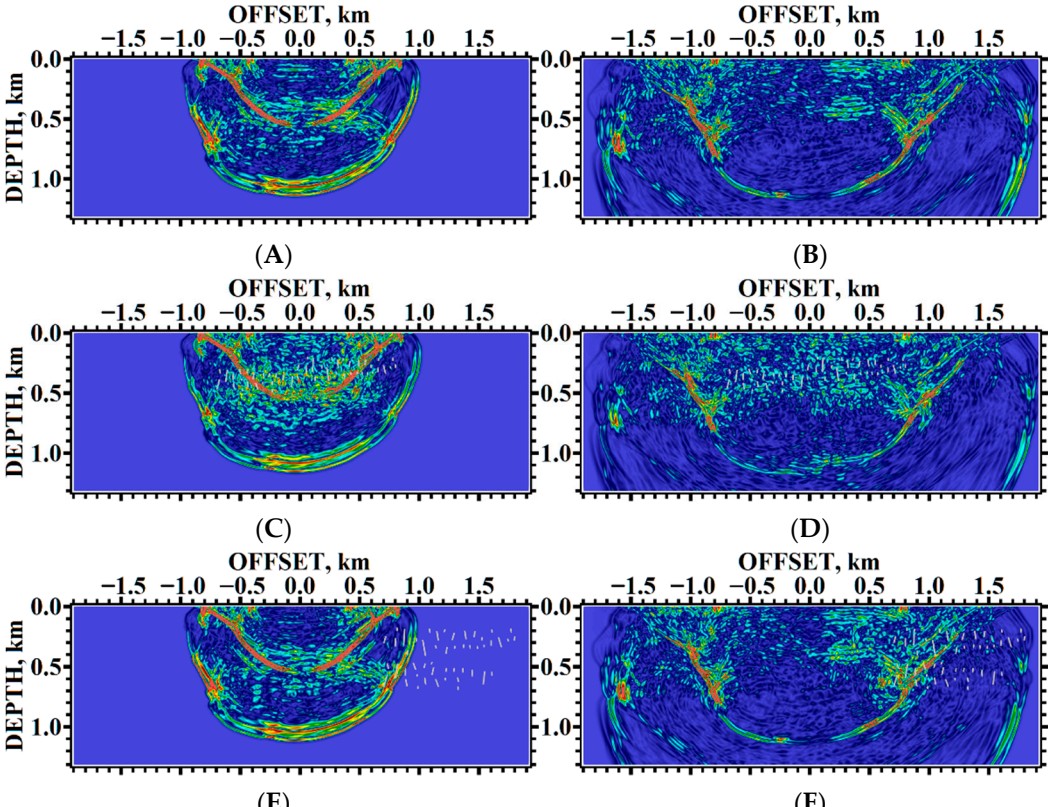

**Figure 33.** Example #5, velocity modulus snapshot: (**A,B**) «without fractures», time of 0.275 s (**A**) and time of 0.51 (**B**); (**C,D**) «fractured zone #1», time of 0.2784925 s (**C**) and time of 0.516477 (**D**); (**E,F**) «fractured zone #2», time of 0.2695 s (**E**) and time of 0.4998 s (**F**).

All scales of fields used in this section are introduced in Figure 31B–D.

The source was a Ricker wavelet with a frequency of 30 Hz, marked by the red dot in Figures 31 and 32A. The space step was 1.6 m. Due to the different geometry of the fractures in the problems «fractured zone #1» and «fractured zone #2», the time steps were different in accordance with the stability conditions. Accordingly, we obtain time steps of 0.2 ms, 0.0779 ms, and 0.07 ms for the problems «without fractured zone», «fractured zone #1», and «fractured zone #2», respectively. Thus, the total time is also slightly different, 1 s, 1.0127 s, 0.98 s, respectively.

For clarity, we have chosen time points 0.275 s and 0.51 s (exact values are given for the problem «without fractured zone») to compare the wave fields. The moment of 0.275 s well reflects the shielding process («fractured zone #1»), and the dynamics of shielding evolution are visible at the time of 0.51 s. For comparison, at 0.275 s («fractured zone #2»), there is practically no shielding since, for this fractured zone and the source position, the re-reflection process begins around the time of 0.51 s.

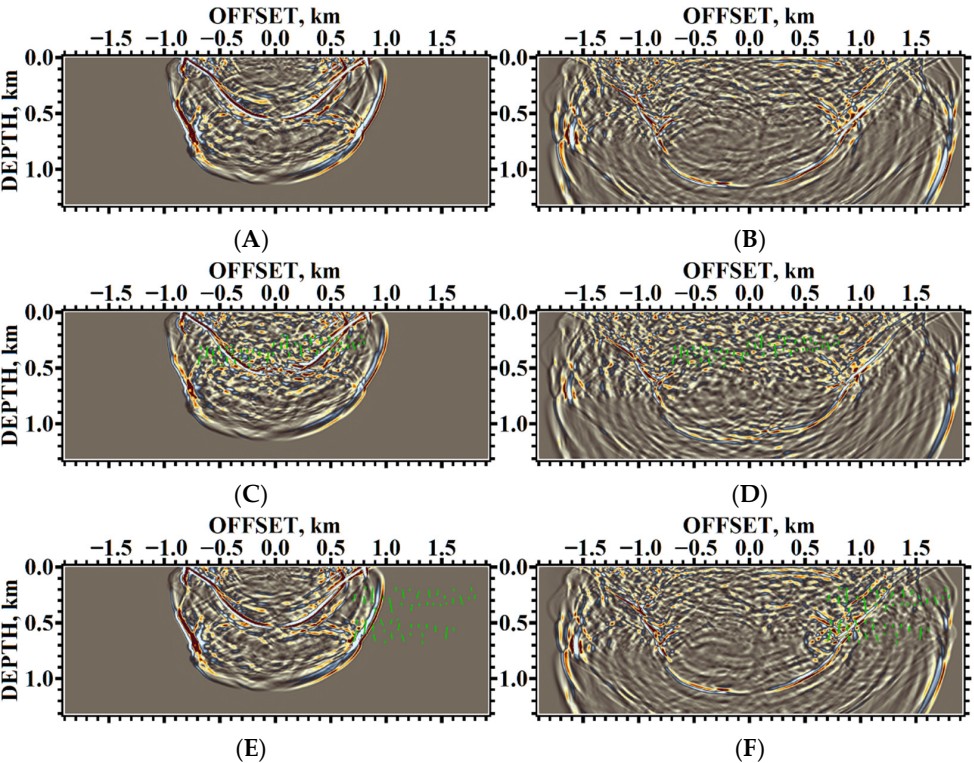

**Figure 34.** Example #5, horizontal component of velocity snapshot: (**A**,**B**) «without fractures», time of 0.275 s (**A**) and time of 0.51 (**B**); (**C**,**D**) «fractured zone #1», time of 0.2784925 s (**C**) and time of 0.516477 (**D**); (**E**,**F**) «fractured zone #2», time of 0.2695 s (**E**) and time of 0.4998 s (**F**).

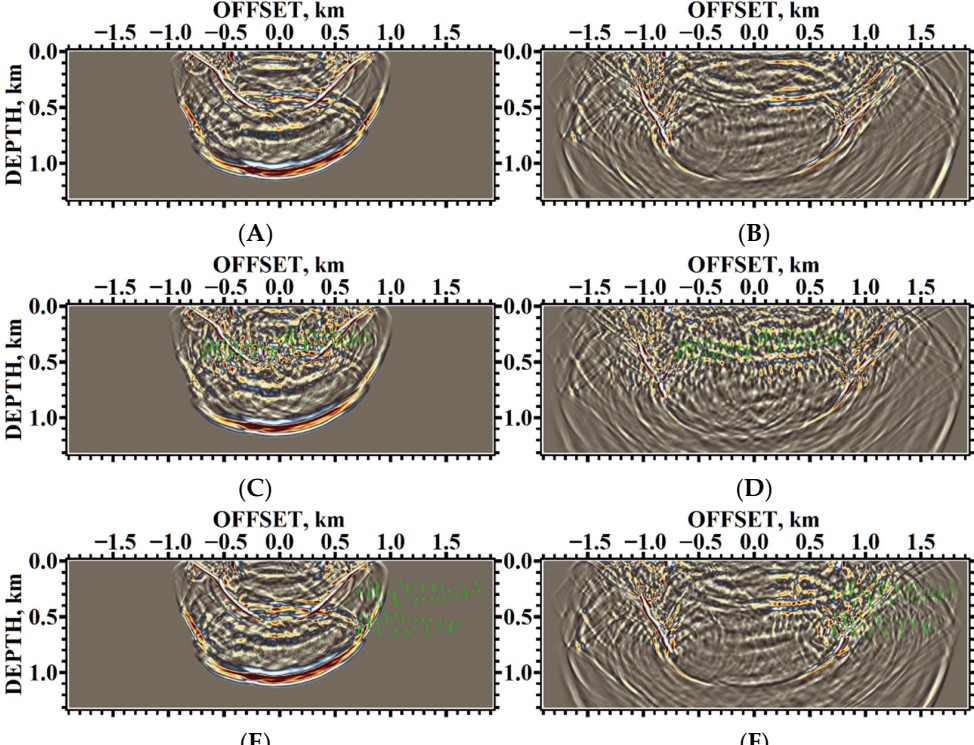

**Figure 35.** Example #5, vertical component of velocity snapshot: (**A**,**B**) «without fractures», time of 0.275 s (**A**) and time of 0.51 (**B**); (**C**,**D**) «fractured zone #1», time of 0.2784925 s (**C**) and time of 0.516477 (**D**); (**E**,**F**) «fractured zone #2», time of 0.2695 s (**E**) and time of 0.4998 s (**F**).

Figures 33–35 show how the shielding process is proceeding. The synthetic seismograms demonstrate visible layers in Figures 36A, 37A and 38A and various shielding options in Figures 36B, 37B and 38B, and Figures 36C, 37C and 38C, respectively.

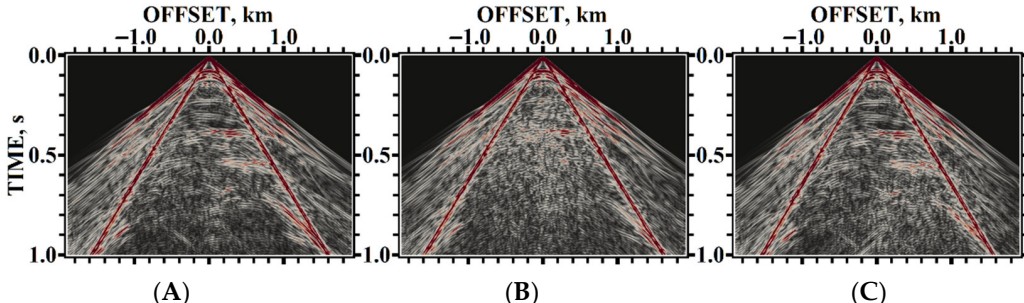

**Figure 36.** Example #5, seismograms, velocity modulus: (**A**) «without fractures»; (**B**) «fractured zone #1»; (**C**) «fractured zone #2».

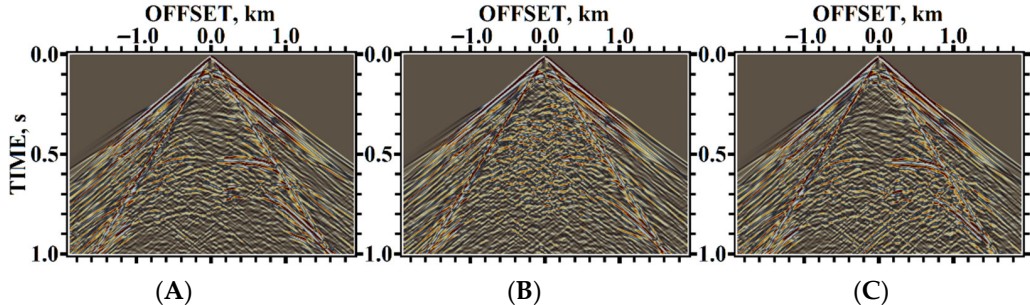

**Figure 37.** Example #5, seismograms, vertical component of velocity: (**A**) «without fractures»; (**B**) «fractured zone #1»; (**C**) «fractured zone #2».

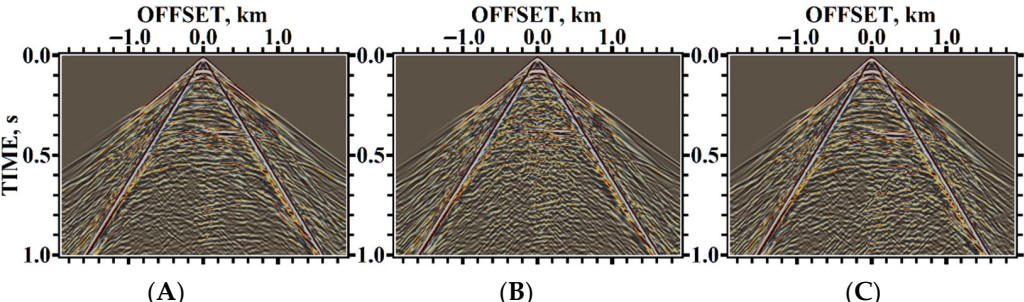

**Figure 38.** Example #5, seismograms, vertical component of velocity: (**A**) «without fractures»; (**B**) «fractured zone #1»; (**C**) «fractured zone #2».

It can be seen that when the fractured zone is located above the metal ore body, the fractured zone has significant shielding properties. While the location of the fractured zone next to the metal ore body, in the case of positioning the source strictly above the metal ore body, although the fractured zone does not provide a significant shielding effect, it also contributes to wave fields and seismograms.

The performed studies show the importance of correct treatment of the scattering of seismic waves on fractured zones for more accurate and precise seismic exploration of minerals.

## 6. Conclusions

In this work, we proposed a novel modification of the numerical grid-characteristic method using overlapping curvilinear meshes. The proposed approach makes it possible to use a Cartesian background computational mesh in the entire integration domain to

accurately simulate the scattering of elastic waves on fractures, which are not codirectional to the coordinate axes, and to avoid interpolation between the overlapping mesh and the background mesh.

The disadvantage of the developed method is that it is not suitable for all configurations of fracture clusters since overlapping meshes must not intersect. However, in nature, most of the fractured zones are characterized by the sub-vertical position of the fractures; therefore, the developed method is applicable in practice.

In contrast to the approach proposed by us in [22], the computational method from this work has a higher accuracy in calculating scattered waves for different fracture orientations. However, the set of geological models to which the method on the overlapping curvilinear mesh is applicable is more limited compared to the use of Chimera meshes. It can also be noted that using curvilinear meshes in the case of fracture's inclination angles close to 45° requires a significant reduction in the time step in accordance with the stability condition.

In terms of direction for further research, we suggest the three-dimensional implementation of the proposed computational method, the development of parallel algorithms, and the application of the proposed method to solve inverse problems [59–61] in geophysics.

**Author Contributions:** Conceptualization, N.I.K. and A.F.; Methodology, N.I.K. and A.F.; Software, N.I.K., A.F. and V.F.; Validation, A.F. and V.F.; Formal analysis, A.F.; Resources, V.F.; Data curation, N.I.K. and A.F.; Writing—original draft, A.F.; Writing—review & editing, N.I.K.; Visualization, A.F.; Supervision, N.I.K.; Project administration, N.I.K. and A.F. All authors have read and agreed to the published version of the manuscript.

**Funding:** This work was conducted with the financial support of the Russian Science Foundation (project No. 20-71-10028).

**Acknowledgments:** This work has been carried out using computing resources of the federal collective usage center Complex for Simulation and Data Processing for Mega-science Facilities at NRC "Kurchatov Institute", http://ckp.nrcki.ru/.

**Conflicts of Interest:** The authors declare no conflict of interest.

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
