# Peer review of "Grid-Characteristic Method on Overlapping Curvilinear Meshes for Modeling Elastic Waves Scattering on Geological Fractures"

_minerals, doi:10.3390/min12121597_

Round 1
Reviewer 1 Report
In this paper, the grid-characteristic method on overlapping curvilinear meshes was proposed to model elastic wave scattering on geological fractures, which is applicable in most cases. The paper is well organized. However, it has room for improvement in writing and graphics.
1. Please indicate that the mathematical physical meaning of each variable appears in equations 2.1.1-2.1.5, such as σ.
2. The number of the Computational algorithm should be 2.2
3. For the 2.2 Computational algorithms, it is suggested to add pseudo code or flow chart to understand the algorithm process more clearly.
4. The calculation strategy described in 2.2 needs to use the content of the next section. It is recommended to change the writing order.
5. Line 157 ' Here ' font is too large, detect a similar error in this paper.
6. The marks A, B, and C in Figure 5.1.1 are not clear.
7. Please check whether the sentence in line 221 is complete. This problem also appears in line 230, etc. Check for similar errors in this paper.
8. There is some problems with typesetting appearing in lines 336 and 337, and similar problems appear in lines 341 and 342.
Author Response
We are deeply grateful to the respected reviewers for the work done and the comments made. We have modified the paper accordingly to these comments.
In this paper, the grid-characteristic method on overlapping curvilinear meshes was proposed to model elastic wave scattering on geological fractures, which is applicable in most cases. The paper is well organized. However, it has room for improvement in writing and graphics.
1. Please indicate that the mathematical physical meaning of each variable appears in equations 2.1.1-2.1.5, such as σ.
Thank you very much. We added relevant definitions in the text.
2. The number of the Computational algorithm should be 2.2
Thank you very much. This was corrected.
3. For the 2.2 Computational algorithms, it is suggested to add pseudo code or flow chart to understand the algorithm process more clearly.
Thank you very much. We added the appropriate pseudo code into the manuscript.
4. The calculation strategy described in 2.2 needs to use the content of the next section. It is recommended to change the writing order.
Thank you very much. We changed the order of subsections.
5. Line 157 ' Here ' font is too large, detect a similar error in this paper.
6. The marks A, B, and C in Figure 5.1.1 are not clear.
7. Please check whether the sentence in line 221 is complete. This problem also appears in line 230, etc. Check for similar errors in this paper.
8. There is some problems with typesetting appearing in lines 336 and 337, and similar problems appear in lines 341 and 342.
Thank you very much. This was corrected.
Reviewer 2 Report
This paper introduced a method for modeling elastic wave scattering on geological fractures, I am unable to evaluate the theory part, but I have the following questions about how to evaluate the modeling results:
1. I am not sure if the paper falls into the scope of Minerals, although it has "geological fractures" in the title, the contents are not related to any "Minerals" problem.
2. Figure 3.4 should provide more details and explanations.
3. All the figures from 3.4 to 5.5.2, it is hard to evaluate, more accurately? even 4.5 to 4.7 have the difference, how do you say they are related to different methods? if you use different gird sizes, it may cause differences.
4. I suggest you can image the forward shot data with RTM and the Least square migration method, and use this to evaluate the accuracy of your method.
5. I also suggest you can build a complex model related to “Minerals”
6. All the figures should have an x label and a y label.
Author Response
We are deeply grateful to the respected reviewers for the work done and the comments made. We have modified the paper accordingly to these comments.
This paper introduced a method for modeling elastic wave scattering on geological fractures, I am unable to evaluate the theory part, but I have the following questions about how to evaluate the modeling results:
1. I am not sure if the paper falls into the scope of Minerals, although it has "geological fractures" in the title, the contents are not related to any "Minerals" problem.
For clarity, we added appropriate text and perform new calculations demonstrating the importance of the developed numerical method for mineral exploration.
2. Figure 3.4 should provide more details and explanations.
Thank you very much. We added more details and explanations at the Figure 3.4 and in the text.
3. All the figures from 3.4 to 5.5.2, it is hard to evaluate, more accurately? even 4.5 to 4.7 have the difference, how do you say they are related to different methods? if you use different gird sizes, it may cause differences.
We specifically used such a discretization, when from 17 to 25 points (in dependence on the inclination between wave front to the coordinate axis) of the background computational mesh are used to treat the P-waves, and from 10 to 15 points are used to treat the S-waves. This allows to more representative compare of the proposed method with available alternatives. We have added relevant explanations to the manuscript. In order to clarify how specifically the Figures 4.5 – 4.7 differ, we have added a series of one-dimensional plots to the manuscript.
4. I suggest you can image the forward shot data with RTM and the Least square migration method, and use this to evaluate the accuracy of your method.
Unfortunately, at the current moment, we do not have the technical ability to quickly make an RTM or LSM migration. Also, we note, that the manuscript is devoted to the fundamental development of numerical methods for more accurate identification of geological heterogeneities in direct seismic modeling. The considered approach can be used in modeling a wide range of problems related to seismic exploration of ore deposits, hydrocarbon deposits, etc. It also allows one to create more accurate mathematical models of geological media. I.e., the manuscript is devoted to direct seismic modeling, and problems with seismic data processing are beyond its scope.
5. I also suggest you can build a complex model related to “Minerals”
Thank you very much. We built this model, performed the calculations and added to the manuscript.
6. All the figures should have an x label and a y label.
Thank you very much. This was corrected.
Round 2
Reviewer 1 Report
b of addressing the reviews. It is suggested to consider the publication of the manuscript.
Reviewer 2 Report
Thanks to the authors answering my suggestions, I think is it fine now, I don't have more suggestions.